# Comparative Study of Antioxidant Potential of Selected Dietary Vitamins; Computational Insights

**DOI:** 10.3390/molecules24091646

**Published:** 2019-04-26

**Authors:** Dinesh R. Pandithavidana, Samith B. Jayawardana

**Affiliations:** Department of Chemistry, Faculty of Science, University of Kelaniya, Kelaniya 11600, Sri Lanka; samjithuok@gmail.com

**Keywords:** antioxidant, HAT, SET–PT, SPLET, BDE, IP, PDE, PA, ETE

## Abstract

Density functional theory (DFT) was used to explore the antioxidant properties of some naturally occurring dietary vitamins, and the reaction enthalpies related to various mechanisms of primary antioxidant action, i.e., hydrogen atom transfer, single electron transfer–proton transfer, and sequential proton loss–electron transfer were discussed in detail. B3LYP, M05-2X, and M06-2X functionals were utilized in this work. For aqueous phase studies, the integral equation formalism polarized continuum model (IEF–PCM) was employed. From the outcomes, hydrogen atom transfer (HAT) was the most probable mechanism for the antioxidant action of this class of compounds. Comparison of found results with experimental data (available in literature), vitamin C possesses the lowest enthalpy values for both proton affinity (PA) and bond dissociation energy (BDE)in the aqueous phase, suggesting it as the most promising candidate as an antioxidant. Accordingly, these computational insights encourage the design of structurally novel, simple vitamins which will be more economical and beneficial in the pharmaceutical industry.

## 1. Introduction

The origin of the harmful process called oxidative stress lies in the extreme production of free radicals such as reactive oxygen species (ROS),reactive nitrogen species (RNS), and reactive sulfur species (RSS) with half-lives of only a few nanoseconds, whose effects can seriously alter cell structures (e.g., membranes) and damage biomolecules such as lipids, lipoproteins, proteins, and nucleic acids [1,2,3]. Among various types of antioxidants, natural vitamins represent a wide group of chemically distinct, water-soluble, and biologically active compounds which serve to inhibit or delay the oxidation of important macromolecules of cells by scavenging those free radicals [4,5].These antioxidants may also have potential to protect against a number of disease conditions such as aging, atherosclerosis, cancer, autoimmune conditions, asthma, and arthritis. Some vitamins (vitamin C and vitamin B6) have been often been reported as antioxidants which have the capability to limit the oxidative damage in humans and lower the risk of certain chronic diseases [6,7,8].

It is broadly recognized that the most vital structural characteristic which facilitates effective antioxidant activity is the presence of one or more conjugated OH groups or COOH groups, which boosts the ability of such a molecule to quench the free radicals [4,5]. There are three major proposed mechanisms which can be used to clarify how antioxidants release the atomic hydrogen from their OH group or COOH group and scavenge free radicals; (1) the hydrogen atom transfer (HAT) mechanism,(2) the single electron transfer–proton transfer (SET–PT) mechanism, and (3) the sequential proton loss–electron transfer (SPLET) mechanism [6,9].Though bond dissociation energy (BDE) is used as a key factor to determine antioxidant capacity of the HAT mechanism, adiabatic ionization potential (IP) and proton dissociation enthalpy (PDE) are used to determine antioxidant efficiency of the SET–PT mechanism, while proton affinity (PA) and electron transfer enthalpy (ETE) are used as computational parameters to the investigate the antioxidant efficiency of the SPLET mechanism [10,11,12] as illustrated in Scheme 1. It should be apparent that all these mechanisms may or may not co-exist, and they are conditional upon the radical characters and solvent properties. Despite this, the net upshot of all these mechanisms is almost the same.

Although some computational studies have been carried out to investigate the antioxidant activity of individual vitamins and their derivatives [6,7], a comparative density functional theory (DFT) study of a group of dietary vitamins (in Figure 1) was not computed. In this investigation, the radical scavenging activity of vitamin A (retinol), vitamin B1 (thiamine), vitamin B3 (nicotinic-d4 acid), vitamin B6 (pyridoxine), and vitamin C (l-ascorbic acid) have been interpreted in terms of some key thermochemical parameters such as BDE, IP, PDE, PA, and ETE. Revelation of the relationship between structure and reactivity is another goal of this study. Efforts have been made to find out such correlation from these thermochemical parameters. Thus, the current investigation may be helpful in shed more light on this front.

## 2. Results and Discussion

To validate the computational procedure, a density functional theory (DFT) method with B3LYP [13], M05-2X [14], and M06-2X [15] functionals were employed to compute enthalpies of BDE, IP, PDE, PA, and ETE of vitamin A, vitaminB1, vitaminB3, vitaminB6, and vitamin C. When M05-2X and M06-2X functionals were used, the BDE, IP, PDE, PA, and ETE data were very close in comparison to the B3LYP functional as illustrated in Table 1 (Appendix A are also available online). Although thermodynamic properties computed from the B3LYP functional had deviated slightly, it was found that “growing orders” of these properties were identical for all three functionals. Thus, mainly results of M05-2X and M06-2Xfunctionalshave been discussed in the further text.

Since these vitamin molecules exhibit their antioxidant potential in an aqueous cellular environment in reality [5,16], optimized structures of vitamins in aqueous phase (water) have been used for the computational investigations as they are shown in Figure 2.

The comparison of bond dissociation enthalpies (BDE) of these dietary vitamins in aqueous phase is shown in Figure 3. The -OH bond dissociation enthalpies grew in this order: Vitamin C < vitamin A < vitamin B6 < vitamin B1 < vitamin-B3.

As BDE is inversely proportional to the antioxidant activity of the compound, having the lowest value for BDE, vitamin C exhibited the highest antioxidant potential through the HAT mechanism. The presence of a conjugated vinylic hydroxyl group in vitamin C favors the formation of an alkoxy radical (RO^●^) which is stabilized through the resonance effect.

It could be assumed that the electron-withdrawing effect and dimer formation effect (it was computed that formation of strong intermolecular hydrogen bonding is thermodynamically more favorable) of the carboxylic (-COOH) group in vitaminB3 may have caused the highest BDE value (and showing the lowest antioxidant potential) when compared to other vitamins. Due to the presence of allylic or vinylic -OH groups, vitamin A and vitaminB6 demonstrated some moderate effect.

One of the numerical parameters associated with the SET-PT mechanism is the ionization potential (IP). Data in Figure 4 indicate how the IPs varies between these dietary vitamins in aqueous phase. It was revealed that vitamin A should be the best electron donor among the studied vitamins. Due to possessing the lowest IP value, vitamin A illustrates the best antioxidant activity according to the first step of the SET-PT mechanism.

Aqueous phase ionization potentials increased in the following order: Vitamin A < vitamin C < vitamin B6 < vitamin B3 < vitamin B1.

Again, the highest IPs were computed for vitaminB1and vitaminB3 due to the existence of a positive charge on vitaminB1and the presence of an electron-withdrawing COOH groupinvitaminB3. The stepwise mechanism of SET–PT is allied to the predisposition of antioxidants in giving electrons. In the case of this mechanism, an electron is transferred from the antioxidant to the free radical leading to the formation of the radical cation ROH^+●^ which consequently deprotonates. Therefore, adiabatic ionization potential (IP) and proton dissociation enthalpy (PDE) are the most significant thermodynamic properties for analyzing the feasibility of this mechanism. The computed proton dissociation enthalpies (PDEs) corresponding to the second step of SET–PT mechanism followed a somewhat different order than the IPs (as shown in Figure 5): Vitamin B1< vitamin C< vitamin B6 < vitamin B3 < vitamin A.

The amount of energy required to form the radical cation is significantly lower than that needed to accomplish the second step of the SET-PT mechanism [11,12]. Accordingly, the deprotonation of the radical cation ROH^+●^ is the most apparent step that limits the reaction rate of such a mechanism in aqueous medium (polar solvents).In general, IPs and PDEs often demonstrate nearly “mirror trends” as the lower the IP, the higher the PDE [17,18].Thus, computed data reveal that the IPs and PDEs of vitaminB1 and vitamin A are mirror trends of each other. Vitamins with low adiabatic ionization potential (IP) values are more susceptible to ionization and have a stronger antioxidant potential. Anyhow, the energies required to accomplish the whole SET–PT mechanism propose that vitaminB1 and vitaminB3 have relatively low antioxidant activity.

When the “combine effect” was considered (as the SET–PT mechanism = IP + PDE), vitamin C exhibited the most favorable antioxidant activity, and vitaminB3was thermodynamically unfavorable to follow the SET–PT mechanism (Table 1).

Deprotonation of the -OH group or -COOH group represents the first step of the SPLET mechanism. The acidity, the nature of hydrogen bonding, and the presence of electron-withdrawing groups are key influences which determine the enthalpy of proton affinity (PA). According to computed results, proton affinities of these selected vitamins shown in Figure 6 increase in this order: Vitamin C < vitamin B3 < vitamin B6 < vitamin B1 < vitamin A.

It could be assumed that the presence of the vinylic conjugated -OH group in vitamin C and the acidic -COOH group in vitaminB3 lowered proton affinities (PA) of these two vitamins in comparison to the rest. The development of the alkoxide anion in the first step of the SPLET mechanism (PA values of Figure 6) requires much more energy to occur than that required in the electron transfer from the alkoxide anion to the free radical (ETEs of Figure 7), showing that the first step must be the slowest one for this SPLET mechanism

To complete the second step of SPLET mechanism, electron transfer should take place. Electron transfer enthalpies (ETEs) of the vitamins were ascending in this order: Vitamin A < vitamin B6 < vitamin B1 < vitamin C < vitamin B3.

In the case of vitamin A, its higher PA is accompanied with considerably lower ETE in comparison to the rest of the studied vitamins. Hence, the deprotonated form of vitamin A should be the finest electron donor from a thermodynamic point of view. Due to the presence of electron-withdrawing carboxylate anions, vitaminB3 exhibited the highest ETE in aqueous phase. Since the SPLET mechanism is the net effect of PA and ETE, vitamin C was again confirmed to possess the highest antioxidant potential when compared to the other dietary vitamins studied here.

## 3. Computational Details

All DFT calculations have been performed using the Gaussian 09 program package (University of Colombo, Department of Chemistry, Colombo-03, Sri Lanka). The geometry of all vitamins, including their radicals, radical cations, and anions, have been fully optimized by employing the hybrid functional of B3LYP, M05-2X, and M06-2X with the basis set of 6-311++G(d, p) (without any constraints) [19]. Optimized structures were confirmed to be real minima by frequency analyses. Aqueous phase data were obtained for geometries of studied optimized species, employing integral equation formalism polarized continuum model (IEF–PCM) [20]. The following thermodynamic properties were computed at T = 298.15 K with non-scaled zero-point energies (ZPE).BDE, IP, PDE, PA, and ETE have been computed in water at 298.15 K following the formulae already applied on similar molecules [20,21], according to the following expressions:BDE =H(RO^●^) + H(H^●^) − H(ROH)(1)
IP = H(ROH^●+^) + H(e^−^) −H(ROH)(2)
PDE= H (RO^●^) + H(H^+^) −H(ROH^●+^)(3)
PA = H(RO^−^) + H(H^+^) −H(ROH)(4)
ETE = H(RO^●^) + H(e^−^) −H(RO^−^)(5)

H(ROH), H(RO^●^), and H(ROH^●+^) represent the protonated form, the radical form, and the radical cation form of the vitamin molecule, respectively. The calculated H(H^+^) and H(e^−^) enthalpies were obtained as 6.197 kJmol^−1^ and 3.145 kJmol^−1^, respectively from the literature [21] available.

## 4. Conclusions

In this computational investigation, antioxidant potentials of the five natural dietary vitamins were explored from the thermodynamic point of view. HAT, SET–PT, and SPLET mechanisms were used to illustrate their radical scavenging activities in the aqueous phase. Employed functionals provided aqueous phase reaction enthalpies in very good mutual agreement. However, results demonstrated that the enthalpies from the three considered mechanisms do have some deviation, which may provide a primary clue to which mechanism is favored over the other.

In water (aqueous phase), deprotonation of the OH or COOH group represented the preferred process. Employing the IEF–PCM model as an implicit solvent model, vitaminB3 and vitaminB1 displayed lower radical scavenging/antioxidant capabilities according to all three mechanisms employed. This was more pronounced in the mechanisms of HAT and SPLET, having relatively low energy pathways when compared to the SET–PT mechanism. However, in some cases, the HAT mechanism (described by BDE) was more applicable than both SET–PT and SPLET mechanisms. This is because smaller values of BDE were more pronounced in the HAT mechanism when compared to relatively higher IP and PA values. Thus, among the selected dietary vitamin molecules, vitamin C has the lowest values for both PA and BDE in the aqueous phase. In general, results of computational investigation positively agreed with the available experimental work, suggesting that vitamin C possesses the highest antioxidant effect in the aqueous environment. It can be concluded that modification of the chemical structure of vitamin C provides insight into the design of structurally novel, simple vitamins which will be more economical and beneficial in pharmaceutical industry.

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
