# Peer review of "Comparative Study of Antioxidant Potential of Selected Dietary Vitamins; Computational Insights"

_molecules, 2019, doi:10.3390/molecules24091646_

Round 1

Reviewer 1 Report

The work deals with an interesting computational study regarding the antioxidant properties of some dietary vitamins. The work is very interesting and well introduced. Unfortunately, I have seen some point to be clarified in the part of data discussion. For this, the work needs a MINOR REVISION before being considered for publication.

In Figures 3-7, all data are shown without a vision of the data variability, for example by showing a standard deviation of the measurements. If possible, please add a study of the result variability to make all the work more clear. Or, alternatively, it could be useful to compare the data according to the average of the three used functionals. In this case, the discussion of the averages could give some slight changes in the rankings with respect to those given by the Authors.

- Lines 8-19: please, also in the Abstract the abbreviations should be explained (as it has been made in the Introduction), otherwise a list of them should be followed soon after and before the body text of the manuscript.

- Lines 77-78: please, better clarify the reason to prefer the M05-2X and M06-2X functionals as the main discussed in the paper and not, as suggested, the average values of all the three used functionals.

- Lines 80-82: the quality of the Figure 2 should be ameliorated, especially for the contrast of the imagine.

- Line 97: it seems that, from Figure 3, vitamin B6 has the second lower value of BDE as referred to the B3LYP and quite equal values between vitamin A and B6 as referred to M05-2X and M06-2X. The ranking order is that vitamin A is the second in the lower BDE value (line 85). Please, explain.

- Lines 105-108: in Figure 4, it seems that data of IP  between vitamin B6 and C seem very similar in all three functionals, but the ranking was vit C < vit B6. Is it exact? Keeping into account the natural variability of the measures, could it be inferred that vit C and B6 have the same IP value? Please, clarify.

- Line 109: correct "B1and" to "B1 and".

- Lines 149-151: as for the SPLET mechanism, I see from Figure 7 that vit B1 and B6 show very similar values and the ranking reported in Line 151 was B6 < B1, where the only clear lower value seems for the B3LYP functional. Please, explain this point, also given the affirmation made in lines 77-78.

Author Response

Response to Reviewer 1 Comments

Point 1: In Figures 3-7, all data are shown without a vision of the data variability, for example by showing a standard deviation of the measurements. If possible, please add a study of the result variability to make all the work more clear. Or, alternatively, it could be useful to compare the data according to the average of the three used functionals. In this case, the discussion of the averages could give some slight changes in the rankings with respect to those given by the Authors.

Lines 77-78: please, better clarify the reason to prefer the M05-2X and M06-2X functionals as the main discussed in the paper and not, as suggested, the average values of all the three used functionals.

Response 1: To validate the computational procedure, density Functional Theory (DFT) method with B3LYP, M05-2X and M06-2X functional were employed. Since data were very close (standard deviation is very less) when M05-2X and M06-2X functionals were used with compared to the B3LYP functional, average of data related to M05-2X and M06-2X functional were used to discuss.

**As authors we really appreciate this valuable point suggested by the reviewer. So according to your suggestion, data of these 3 major mechanisms have been arranged/presented in this manner (including standard deviations)

(1)   HAT mechanism= BDE

(2)   SET-PT mechanism= IP + PDE

(3)   SPLET mechanism= PA + ETE

Table 1. Reaction enthalpies in kJmol-1 in water for the selected dietary vitamins, employing mean functional values of M05-2X and M06-2X

Vitamins

HAT mechanism

(BDE)

SET-PT mechanism

(IP   + PDE)

SPLET mechanism

(PA   + ETE)

Vitamin A

356.0   (±1.0)

1742.0   (±2.0)

532.0   (±1.0)

Vitamin B1

435.0 (±0.0)

1750.5 (±0.5)

611.5 (±1.5)

Vitamin B3

481.5 (±0.5)

1790.5 (±1.5)

658.0 (±2.0)

Vitamin B6

360.5   (±1.5)

1655.0   (±1.0)

537.0   (±0.0)

Vitamin C

318.5   (±0.5)

1611.5   (±1.5)

495.0   (±2.0)

Point 2: Lines 8-19: please, also in the Abstract the abbreviations should be explained (as it has been made in the Introduction), otherwise a list of them should be followed soon after and before the body text of the manuscript.

Response 2: Yes we did the modification as you mentioned.

Comparison of found results with experimental data (available in literature), vitamin-C possesses the lowest values for both proton affinity (PA) and bond dissociation enthalpy (BDE) at the aqueous phase, exhibiting that most promising candidate as an antioxidant.

Point 3: Lines 80-82: the quality of the Figure 2 should be ameliorated, especially for the contrast of the imagine.

Response 3: yes we tried to improve the resolution as much as possible in new version.

Point 4: Line 97: it seems that, from Figure 3, vitamin B6 has the second lower value of BDE as referred to the B3LYP and quite equal values between vitamin A and B6 as referred to M05-2X and M06-2X. The ranking order is that vitamin A is the second in the lower BDE value (line 85). Please, explain.

Response 4: As we mentioned earlier, computed results of B3LYP exhibited unusual deviation from the mean value. That’s why we considered data generated from M05-2X and M06-2X to discuss our findings. To just validate the procedure, we employed all three functional as mentioned in literature.

ü  Skorna, P.; Rimarcík, J.; Poliak, P.; Lukes, V.; Klein, E. Thermodynamic study of vitamin B6 antioxidant potential,Computational and Theoretical Chemistry2016, 1077, 32–38

ü  Bendich, A.; Machlin, L.J.; Scandurra, O.; Burton, G.W.; Wayner, D.D.M. The antioxidant role of vitamin C.Advances in Free Radical Biology & Medicine1986, 2(2), 419-444

ü  Mazzone, G.; Russo, N.; Toscano, M. Antioxidant properties comparative study of natural hydroxycinnamic acids and structurally modified derivatives: Computational insights. Computational and Theoretical Chemistry2016, 1077,39–47.

Point 5: Lines 105-108: in Figure 4, it seems that data of IP  between vitamin B6 and C seem very similar in all three functionals, but the ranking was vit C < vit B6. Is it exact? Keeping into account the natural variability of the measures, could it be inferred that vit C and B6 have the same IP value? Please, clarify.

Response 5: Comparing IP value alone, we may not the right picture of the antioxidant potentials of these vitamins because SET-PT mechanism is a total effect of both IP + PDE. This was our mistake as considering IP and PDE separately. As we mentioned in “Response-1; data table” we should consider the net effect of IP + PDE. According to that SET-PT mechanism is more favourable in vitamin-C and vitamin-B6.

Point 6: Line 109: correct "B1and" to "B1 and".

Response 6: space detection was changed due to the version of MS word.

Point 7: Lines 149-151: as for the SPLET mechanism, I see from Figure 7 that vit B1 and B6 show very similar values and the ranking reported in Line 151 was B6 < B1, where the only clear lower value seems for the B3LYP functional. Please, explain this point, also given the affirmation made in lines 77-78.

Response 7:As you pointed out in Figure 7; although vitamin B1 and B6 show almost similar values for ETE calculations, but to get the correct picture, we have to consider the total effect of both PA plus ETE. (since SPLET mechanism= PA + ETE). It was our mistake and we will correct it on manuscript.  As we mentioned earlier, computed results of B3LYP exhibited unusual deviation from the mean value. That’s why we considered data generated from M05-2X and M06-2X to discuss our findings.

Reviewer 2 Report

The authors present computed values of energetic values associated with passage of a generalized alcoholic -OH group to the radical -O· . The hydrogen atom transfer (HAT) process is characterized by a single quantity, the Bond Dissociation Energy (BDE). A Single Electron Transfer (SET) followed by proton loss involves the Ionization potential (IP) and proton dissociation energy (PDE). A proton loss followed by electron loss refers a proton affinity (PA) and to the electron transfer enthalpy (ETE). The authors employ reasonable choices of density functional (Minnesots 05-2X and 06-2X) and a basis 6-311++G(d,p) that seems adequate to a balanced treatment of radicals, anions, and cations. Medium effects are modeled by the IEF-PCM polarizable continuum, parametrized for water. No specific solvation effects are considered.

The authors present graphs comparing the computed parameters listed above, for Vitamins A, B1, B3, B6, and C. For example, The BDE values suggest that Vitamin C (ca 300 kJ/mol) would be more likely to follow the HAT mechanism than any other vitamin; B1 would be least likely 480 kJ/mol). Similar graphs are presented for each quantity. Oddly, the sum of parameters bearing on the SET-PT and SPLET paths are not reported. The results would look like this:

BDE

IP+PDE

PA+ETE

A

355

1750

535

B1

435

1740

605

B3

480

1760

650

B6

360

1660

530

C

320

1610

495

I have estimated values from the charts, since precise numbers are not provided. Considering only these values for each mechanism, it appears that C is most likely of all the vitamins to follow SET-PT path, and is also the most likely to follow the SPLET path.

The authors conclude that the HAT mechanism is favored for every species, and indeed the BDE energy is smaller than the alternatives in every case. However considering only the computed values reported in this MS, one cannot tell which path a particular vitamin might prefer. Isn’t it the case that all three paths lead to the oxy radical? If so, the overall enthalpies for the processes for each vitamin must be identical. The authors assume that the reactions produce solvated electrons and protons but it seems most unlikely that the electron itself would persist without reacting further. One may also wonder how reactions with requirements of hundreds of kJ/mol can happen, but we presume the subsequent reaction of the oxy-radical will be strongly exothermic.

Lesser objections:

I found the sequence of presentation to be unconventional. Usually (once the problem is stated) the details of computation are presented, followed by results and discussion. Here many key features of the computational modeling (basis, solvent model) are deferred.

Are any experimental data available on the antioxidation effects of these vitamins?

Rationale of the high BDE for B3 (alluding to dimerization, lines 91-95)) cannot apply to the computed values, since no consideration of dimerization is included in computations.

Author Response

Response to Reviewer 2 Comments

Point 1: Medium effects are modeled by the IEF-PCM polarizable continuum, parametrized for water. No specific solvation effects are considered.

Response 1: The polarizable continuum model (PCM) is a commonly used method in computational chemistry to model solvation effects. If it were necessary to consider each solvent molecule as a separate molecule, the computational cost of modeling a solvent-mediated chemical reaction would grow prohibitively high.

**As a developing country, our state universities have limited facilities. That’s why we employed modeling the solvent as a “polarizable continuum”, rather than individual molecules, makes ab initio computation feasible.

Please provide your response for Point 1. (in red)

Point 2: The authors present graphs comparing the computed parameters listed above, for Vitamins A, B1, B3, B6, and C. For example, The BDE values suggest that Vitamin C (ca 300 kJ/mol) would be more likely to follow the HAT mechanism than any other vitamin; B1 would be least likely 480 kJ/mol). Similar graphs are presented for each quantity. Oddly, the sum of parameters bearing on the SET-PT and SPLET paths are not reported. The results would look like this

Response 2: As authors we really appreciate this valuable point suggested by the reviewer. So according to your suggestion, data of these 3 major mechanisms have been arranged/presented in this manner (including standard deviations)

(1)     HAT mechanism= BDE

(2)     SET-PT mechanism= IP + PDE

(3)   SPLET mechanism= PA + ETE

Table 1. Reaction enthalpies in kJmol-1 in water for the selected dietary vitamins, employing mean functional values of M05-2X and M06-2X

Vitamins

HAT mechanism

(BDE)

SET-PT mechanism

(IP   + PDE)

SPLET mechanism

(PA   + ETE)

Vitamin A

356.0   (±1.0)

1742.0   (±2.0)

532.0   (±1.0)

Vitamin B1

435.0 (±0.0)

1750.5 (±0.5)

611.5 (±1.5)

Vitamin B3

481.5 (±0.5)

1790.5 (±1.5)

658.0 (±2.0)

Vitamin B6

360.5   (±1.5)

1655.0   (±1.0)

537.0   (±0.0)

Vitamin C

318.5   (±0.5)

1611.5   (±1.5)

495.0   (±2.0)

Point 3: The authors conclude that the HAT mechanism is favored for every species, and indeed the BDE energy is smaller than the alternatives in every case. However considering only the computed values reported in this MS, one cannot tell which path a particular vitamin might prefer. Isn’t it the case that all three paths lead to the oxy radical?

Response 3: As the reviewer suggested, after we consider “combine effect” (SET-PT mechanism= IP + PDE and SPLET mechanism= PA + ETE), conclusions are more straight forward. E.g. Vitamin-C serves as the best antioxidant which may follow either HAT mechanism or SPLET mechanism which are energetically more favourable with compared to the SET-PT mechanism.

Point 4: One may also wonder how reactions with requirements of hundreds of kJ/mol can happen, but we presume the subsequent reaction of the oxy-radical will be strongly exothermic.

Response 4: According to literature, researchers who computed antioxidant activities of vitamin-C and vitamin-B6 also have reported such higher enthalpy values (hundreds of kJ/mol).

Skorna, P.; Rimarcík, J.; Poliak, P.; Lukes, V.; Klein, E. Thermodynamic study of vitamin B6 antioxidant potential, Computational and Theoretical Chemistry2016, 1077, 32–38

Bendich, A.; Machlin, L.J.; Scandurra, O.; Burton, G.W.; Wayner, D.D.M. The antioxidant role of vitamin C. Advances in Free Radical Biology & Medicine1986, 2(2), 419-444.

***The hydrolysis of ATP into ADP and inorganic phosphate releases 30.5 kJ/mol of enthalpy, with a change in free energy of 3.4 kJ/mol. [Gajewski, E.; Steckler, D.; Goldberg, R. (1986). "Thermodynamics of the hydrolysis of adenosine 5′-triphosphate to adenosine 5′-diphosphate" (PDF). J. Biol. Chem. 261 (27): 12733–12737].

So we can assume that since those antoxidative activities take place in a “cellular environment” in reality, 10-15 moles of ATP (305-458 kJ/mol) can be consumed to precede these mechanisms.

** plus according to our experience related to computational studies, this is not a good tool to predict “absolute value / absolute enthalpy” of any molecule or molecular reaction because these models are based on several assumptions and limitations. But we can use this as a powerful tool to “compare trends”( which one is more favourable than  the others) in these type of biological applications.

Point 5: I found the “sequence of presentation” to be unconventional. Usually (once the problem is stated) the details of computation are presented, followed by results and discussion.

Response 5: Yes, we agree with you with compared to the format of “other indexed journals”. But we have followed the template and format of the journal of Molecules (ISSN 1420-3049). That’s how they applied. (Introduction, results & discussion followed by methodology)

Point 6: Here many key features of the computational modeling (basis, solvent model) are deferred.

Response 6: I couldn’t understand why the reviewer pointed out that “computational modelling (basis, solvent model) are deferred” because we employed the “standard computational procedures used by other researchers who worked on same type of anti-oxidant molecules.

Please refer these references:

ü  Skorna, P.; Rimarcík, J.; Poliak, P.; Lukes, V.; Klein, E. Thermodynamic study of vitamin B6 antioxidant potential,Computational and Theoretical Chemistry2016, 1077, 32–38

ü  Bendich, A.; Machlin, L.J.; Scandurra, O.; Burton, G.W.; Wayner, D.D.M. The antioxidant role of vitamin C.Advances in Free Radical Biology & Medicine1986, 2(2), 419-444

ü  McCall, M.R.; Frei, B. Can antioxidant vitamins materially reduce oxidative damage in humans? Free Radic Biol Med. 1999, 26, 1034-1053. doi:10.1016/S0891-5849(98)00302-5

ü  Mazzone, G.; Russo, N.; Toscano, M. Antioxidant properties comparative study of natural hydroxycinnamic acids and structurally modified derivatives: Computational insights. Computational and Theoretical Chemistry2016, 1077,39–47.

Point 7: Are any experimental data available on the antioxidation effects of these vitamins?

Response 7: Yes, some literate is available regarding experimental data on the antioxidation effects of some of these vitamins. (In Electron Spin Resonance (ESR) study of the activity of various vitamins against the radical-mediated oxidative damage

in human whole blood, relatively high antioxidant activity, in comparison

to ascorbic acid (Vitamin C), was observed for vitamin B6).

Ref:

ü  P. Stocker, J.-F. Lesgards, N. Vidal, F. Chalier, M. Prost, ESR study of a biological

assay on whole blood: antioxidant efficiency of various vitamins, Biochim.

Biophys. Acta 793 (2003) 1–8.

ü  Gliszczyn´ ska-S´ wigło, Antioxidant activity of water soluble vitamins in the

TEAC (trolox equivalent antioxidant capacity) and the FRAP (ferric reducing

antioxidant power) assays, Food Chem. 96 (2006) 131–136.

**Those experimental data supported our computational findings.

Point 8: Rationale of the high BDE for B3 (alluding to dimerization, lines 91-95)) cannot apply to the computed values, since no consideration of dimerization is included in computations.

Response 8: It’s a very good point; we missed to include it to the section of discussion. Due to the presences of unusually higher BDE value for Vitamin-B3, we separately modelled (computed) the dimer-formation and its effect on ionization potential. (That part will be included to modified manuscript).

**PDF version of the comments also has been enclosed here..

Round 2

Reviewer 2 Report

Response to Reviewer Comments

 Reviewer  followup

Point 1: Medium effects are modeled by the IEF-PCM polarizable continuum, parametrized for water. No specific solvation effects are considered.

Response 1: The polarizable continuum model (PCM) is a commonly used method in computational chemistry to model solvation effects. If it were necessary to consider each solvent molecule as a separate molecule, the computational cost of modeling a solvent-mediated chemical reaction would grow prohibitively high.

**As a developing country, our state universities have limited facilities. That’s why we employed modeling the solvent as a “polarizable continuum”, rather than individual molecules, makes ab initio computation feasible.

Reviewer reply: We agree that a full molecular description of a condensed systems is beyond reach - this point is not an objection, but merely an observation. Speaking generally, even a single water molecule can have notable effects on reaction energies and pathways. 

Point 2: The authors present graphs comparing the computed parameters listed above, for Vitamins A, B1, B3, B6, and C. For example, The BDE values suggest that Vitamin C (ca 300 kJ/mol) would be more likely to follow the HAT mechanism than any other vitamin; B1 would be least likely 480 kJ/mol). Similar graphs are presented for each quantity. Oddly, the sum of parameters bearing on the SET-PT and SPLET paths are not reported. The results would look like this

Response 2: As authors we really appreciate this valuable point suggested by the reviewer. So according to your suggestion, data of these 3 major mechanisms have been arranged/presented in this manner (including standard deviations)

(1)     HAT mechanism= BDE

(2)     SET-PT mechanism= IP + PDE

(3)   SPLET mechanism= PA + ETE

Table 1. Reaction enthalpies in kJmol-1 in water for the selected dietary vitamins, employing mean functional values of M05-2X and M06-2X

Vitamins

HAT mechanism

(BDE)

SET-PT mechanism

(IP   + PDE)

SPLET mechanism

(PA   + ETE)

Vitamin A

356.0   (±1.0)

1742.0   (±2.0)

532.0   (±1.0)

Vitamin B1

435.0 (±0.0)

1750.5 (±0.5)

611.5 (±1.5)

Vitamin B3

481.5 (±0.5)

1790.5 (±1.5)

658.0 (±2.0)

Vitamin B6

360.5   (±1.5)

1655.0   (±1.0)

537.0   (±0.0)

Vitamin C

318.5   (±0.5)

1611.5   (±1.5)

495.0   (±2.0)

Reviewer reply: Thanks for this constructive response.

Point 3: The authors conclude that the HAT mechanism is favored for every species, and indeed the BDE energy is smaller than the alternatives in every case. However considering only the computed values reported in this MS, one cannot tell which path a particular vitamin might prefer. Isn’t it the case that all three paths lead to the oxy radical?

Response 3: As the reviewer suggested, after we consider “combine effect” (SET-PT mechanism= IP + PDE and SPLET mechanism= PA + ETE), conclusions are more straight forward. E.g. Vitamin-C serves as the best antioxidant which may follow either HAT mechanism or SPLET mechanism which are energetically more favourable with compared to the SET-PT mechanism.

 Reviewer reply: In my first remark I overlooked the fact that the products include species other than the oxy radical. For the path specified by BDE, the accompanying product is H atom (solvated) while for the other paths H(+) and the electron (solvated) appear. For comparison we should trace the reactivity of H atom in one case and the ion pair in the other cases, or else add in the recombination energy by which H atom is re-formed.

Point 4: One may also wonder how reactions with requirements of hundreds of kJ/mol can happen, but we presume the subsequent reaction of the oxy-radical will be strongly exothermic.

Response 4: According to literature, researchers who computed antioxidant activities of vitamin-C and vitamin-B6 also have reported such higher enthalpy values (hundreds of kJ/mol).

Skorna, P.; Rimarcík, J.; Poliak, P.; Lukes, V.; Klein, E. Thermodynamic study of vitamin B6 antioxidant potential, Computational and Theoretical Chemistry2016, 1077, 32–38

Bendich, A.; Machlin, L.J.; Scandurra, O.; Burton, G.W.; Wayner, D.D.M. The antioxidant role of vitamin C. Advances in Free Radical Biology & Medicine1986, 2(2), 419-444.

***The hydrolysis of ATP into ADP and inorganic phosphate releases 30.5 kJ/mol of enthalpy, with a change in free energy of 3.4 kJ/mol. [Gajewski, E.; Steckler, D.; Goldberg, R. (1986). "Thermodynamics of the hydrolysis of adenosine 5′-triphosphate to adenosine 5′-diphosphate" (PDF). J. Biol. Chem. 261 (27): 12733–12737].

So we can assume that since those antoxidative activities take place in a “cellular environment” in reality, 10-15 moles of ATP (305-458 kJ/mol) can be consumed to precede these mechanisms.

** plus according to our experience related to computational studies, this is not a good tool to predict “absolute value / absolute enthalpy” of any molecule or molecular reaction because these models are based on several assumptions and limitations. But we can use this as a powerful tool to “compare trends”( which one is more favourable than  the others) in these type of biological applications.

  Reviewer reply: This response is eminently reasonable.

Point 5: I found the “sequence of presentation” to be unconventional. Usually (once the problem is stated) the details of computation are presented, followed by results and discussion.

Response 5: Yes, we agree with you with compared to the format of “other indexed journals”. But we have followed the template and format of the journal of Molecules (ISSN 1420-3049). That’s how they applied. (Introduction, results & discussion followed by methodology)

  Reviewer reply: Of course writers must follow the journal conventions. My remark also applies to point 6, in which the details of computation appear late in the presentation. That is what I meant by “deferred.” That was not an objection to the choice of methods.

Point 6: Here many key features of the computational modeling (basis, solvent model) are deferred.

Response 6: I couldn’t understand why the reviewer pointed out that “computational modelling (basis, solvent model) are deferred” because we employed the “standard computational procedures used by other researchers who worked on same type of anti-oxidant molecules.

Please refer these references:

ü  Skorna, P.; Rimarcík, J.; Poliak, P.; Lukes, V.; Klein, E. Thermodynamic study of vitamin B6 antioxidant potential,Computational and Theoretical Chemistry2016, 1077, 32–38

ü  Bendich, A.; Machlin, L.J.; Scandurra, O.; Burton, G.W.; Wayner, D.D.M. The antioxidant role of vitamin C.Advances in Free Radical Biology & Medicine1986, 2(2), 419-444

ü  McCall, M.R.; Frei, B. Can antioxidant vitamins materially reduce oxidative damage in humans? Free Radic Biol Med. 1999, 26, 1034-1053. doi:10.1016/S0891-5849(98)00302-5

ü  Mazzone, G.; Russo, N.; Toscano, M. Antioxidant properties comparative study of natural hydroxycinnamic acids and structurally modified derivatives: Computational insights. Computational and Theoretical Chemistry2016, 1077,39–47.

Point 7: Are any experimental data available on the antioxidation effects of these vitamins?

Response 7: Yes, some literat[ur]e is available regarding experimental data on the antioxidation effects of some of these vitamins. (In Electron Spin Resonance (ESR) study of the activity of various vitamins against the radical-mediated oxidative damage

in human whole blood, relatively high antioxidant activity, in comparison

to ascorbic acid (Vitamin C), was observed for vitamin B6).

Ref:

ü  P. Stocker, J.-F. Lesgards, N. Vidal, F. Chalier, M. Prost, ESR study of a biological

assay on whole blood: antioxidant efficiency of various vitamins, Biochim.

Biophys. Acta 793 (2003) 1–8.

ü  Gliszczyn´ ska-S´ wigło, Antioxidant activity of water soluble vitamins in the

TEAC (trolox equivalent antioxidant capacity) and the FRAP (ferric reducing

antioxidant power) assays, Food Chem. 96 (2006) 131–136.

**Those experimental data supported our computational findings.

  Reviewer reply: This is useful, and worth incorporating in the text. I find no mention of the experimental work, and these references do not appear in the citation list.

Point 8: Rationale of the high BDE for B3 (alluding to dimerization, lines 91-95)) cannot apply to the computed values, since no consideration of dimerization is included in computations.

Response 8: It’s a very good point; we missed to include it to the section of discussion. Due to the presences of unusually higher BDE value for Vitamin-B3, we separately modelled (computed) the dimer-formation and its effect on ionization potential. (That part will be included to modified manuscript).

 Reviewer reply: I found a new passage in lines 99-102 referring more explicitly to dimerization. It needs clarification, and numerical values.

Author Response

Reviewer reply: I found a new passage in lines 99-102 referring more explicitly to dimerization. It needs clarification, and numerical values.

Respond to Reviewer’s reply (Point 8):

According to our computational investigations related to “dimer formation”, it was evident that the process is thermodynamically more favorable (having -42.39 kJmol-1) with compared to its antioxidant activities.

For your references; our computational data & procedure are enclosed here for this dimerization.

  Vitamin-B3 

A)     

 Zero-point correction=                           0.103090 (Hartree/Particle)

 Thermal correction to Energy=                    0.110183

 Thermal correction to Enthalpy=                  0.111127

 Thermal correction to Gibbs Free Energy=         0.070875

 Sum of electronic and zero-point Energies=           -436.893373

 Sum of electronic and thermal Energies=              -436.886279

 Sum of electronic and thermal Enthalpies=            -436.885335

 Sum of electronic and thermal Free Energies=         -436.925587

B)   

Zero-point correction=                           0.207423 (Hartree/Particle)

 Thermal correction to Energy=                    0.223001

 Thermal correction to Enthalpy=                  0.223945

 Thermal correction to Gibbs Free Energy=         0.160690

 Sum of electronic and zero-point Energies=           -873.803337

 Sum of electronic and thermal Energies=              -873.787760

 Sum of electronic and thermal Enthalpies=            -873.786816

 Sum of electronic and thermal Free Energies=         -873.850071

Entahlpy

E = B – 2A

E = (-873.786816) – {2(-436.885335)}(Hartree/Particle)

E = - 0.016146 (Hartree/Particle)

E = -42.39132623 kJmol-1

Calculation details:

# opt freq b3lyp/6-311++g(d,p) scrf=(iefpcm,solvent=water)

Energy optimization and Frequency optimization

Method: B3LYP

Basis set: 6-311++g(d,p)

Solvation: IEFPCM

Solvent: water
